# The Effect of Obesity on Short- and Long-Term Outcome after Surgical Treatment for Acute Type A Aortic Dissection

**DOI:** 10.3390/life14080955

**Published:** 2024-07-30

**Authors:** Philipp Pfeiffer, Karen Wittemann, Leon Mattern, Vanessa Buchholz, Hazem El Beyrouti, Ahmed Ghazy, Mehmet Oezkur, Georg Daniel Duerr, Chris Probst, Hendrik Treede, Daniel-Sebastian Dohle

**Affiliations:** Department of Cardiovascular Surgery, University Medical Center Mainz, 55131 Mainz, Germany; karen.wittemann@unimedizin-mainz.de (K.W.); leon.mattern@unimedizin-mainz.de (L.M.); vwirtz@students.uni-mainz.de (V.B.); hbeyrouti@gmail.com (H.E.B.); ahmed.ghazy@unimedizin-mainz.de (A.G.); mehmet.oezkur@unimedizin-mainz.de (M.O.); daniel.duerr@unimedizin-mainz.de (G.D.D.); chris.probst@unimedizin-mainz.de (C.P.); hendrik.treede@unimedizin-mainz.de (H.T.); daniel-sebastian.dohle@unimedizin-mainz.de (D.-S.D.)

**Keywords:** aortic dissection, aortic surgery, obesity, long-term survival

## Abstract

Background: A paradox of lower morbidity and mortality in overweight or obese patients undergoing cardiac surgery has been described; however, knowledge about the influence of obesity in patients with acute Type A aortic dissection (AAD) is limited. This study aimed to evaluate the effect of obesity on short- and long-term outcomes after surgical treatment for AAD. Methods: Between 01/2004 and 12/2022, 912 patients with a BMI of 18.5 or greater were operated on for AAD. Patients were grouped according to their BMI (normal weight: BMI 18.5–24.9, *n* = 332; overweight: BMI 25–29.9, *n* = 367; obesity class I: BMI 30–34.9, *n* = 133; obesity class II+: BMI ≥ 35, *n* = 67), and the obtained clinical and surgical data were compared. Results: Obese patients were younger at the time of AAD (*p* = 0.001) and demonstrated higher rates of typical cardiovascular comorbidities (arterial hypertension, *p* = 0.005; diabetes mellitus, *p* < 0.001). The most important preoperative parameters, as well as the surgical approach, were similar between all four groups. The occurrence of renal failure requiring dialysis was higher in patients with BMI ≥ 35 (*p* = 0.010), but the in-hospital (*p* = 0.461) and long-term survival (*p* = 0.894) showed no significant differences. Conclusions: There are no indications that the obesity paradox is applicable in the setting of AAD. Since obese patients are affected by AAD at a younger age, obesity might constitute a risk factor for AAD. However, obesity does not influence short- or long-term survival. Regardless of body weight, immediate surgical therapy remains the treatment of choice for AAD.

## 1. Introduction

Obesity is an important risk factor for the development of a number of chronic diseases, including diabetes mellitus Type 2, obstructive sleep apnea, hypertension, and dyslipidemia, which in turn are risk factors for cardiovascular diseases such as myocardial infarction (MI), coronary artery disease (CAD), cerebrovascular disease, stroke, and congestive heart failure (CHF) [1]. Despite obesity being an established cardiovascular risk factor, numerous studies have shown that in patients with existing heart failure, being overweight or obese seemingly exerts a protective effect that limits the effects of the disease and lowers mortality from all causes, as well as from cardiac diseases [1,2,3,4,5]. These counterintuitive findings have been termed the “obesity paradox”.

Similar paradoxically better outcomes in overweight or obese patients have been reported in surgical and interventional patient populations for both patients undergoing cardiac and non-cardiac procedures [6,7,8,9,10,11]. Some studies suggest that both underweight and morbidly obese patient populations have the highest mortality rates, suggesting a U-shaped curve of optimal weight for the lowest mortality after surgery [6,7].

In acute Type A aortic dissection (AAD), a tear in the intima of the ascending aorta creates a false lumen. Since this can lead to aortic rupture and death, it is a surgical emergency with a mortality of up to 5% per hour for untreated patients after the initial start of symptoms [12]. Autopsy studies indicate that AAD affects up to 12 per 100,000 persons per year [13,14]. Surgical treatment is guided by the extension of the dissection and the hemodynamic status of the patient, but obesity has no immediate influence on operative treatment [15].

Little is known about the influence of obesity in the setting of surgically treated AAD, as the currently published literature on this topic is limited [16]. Some studies have shown a higher BMI or obesity to be a risk factor for 30-day or in-hospital mortality [17,18,19,20,21], but most studies with a long-term follow-up did not find a significantly different long-term survival [17,18,22]. Other studies did not find any significant differences in postoperative survival but found prolonged ventilation or hospital stay in obese patients [22,23]. While the obesity paradox has not been described in patients with AAD, some publications have observed a slightly better survival for overweight patients [23,24,25]. However, these results were not statistically significant.

With an increasing prevalence of obesity and cardiovascular diseases like arterial hypertension, which is the most common risk factor for AAD, knowledge about the influence of obesity in patients undergoing AAD surgery has gained importance [26,27].

We conducted this retrospective study in order to evaluate whether the body mass index (BMI) influences the outcome of patients undergoing surgical repair for AAD.

## 2. Materials and Methods

### 2.1. Study Design and Patients

All patients who underwent surgical treatment for AAD between 01/2004 and 12/2022 were identified using the institutional aortic database of the Department of Cardiovascular Surgery at the University Medical Center Mainz, Germany. In accordance with the Declaration of Helsinki, the study was approved by the institutional review board of the medical association of the state of Rheinland-Pfalz (2018-13574-Epidemiologie), and informed patient consent was waived due to retrospective data collection.

In total, 924 consecutive cases of surgically treated AAD were identified. After excluding 12 underweight patients (BMI < 18.5), 912 patients were included in this study, and the corresponding data were obtained from clinical databases.

Patients were grouped according to their BMI using widely used obesity classes [28]. Due to the decreasing numbers of patients with class III obesity (BMI ≥ 40), class II and III obese patients were pooled, thus creating four groups: normal weight: BMI 18.5–24.9; overweight: BMI 25–29.9; obesity class I: BMI 30–34.9; obesity class II+: BMI ≥ 35.

The postoperative course, including in-hospital mortality to report short-term survival, and long-term results were subsequently compared between groups. While the primary outcome was short- and long-term survival, a descriptive analysis of postoperative complications was also performed. The analyzed data included possible confounding variables, such as patient demographics, comorbidities, and surgical details.

### 2.2. Clinical and Operative Details

Some details of the surgical procedure or management of AAD varied during the 19-year study period. Nevertheless, a common routine is briefly described. Diagnosis of AAD is generally determined using computed tomography (CT), with angiography and transesophageal or transthoracic echography playing only a minor role in the initial diagnosis. For further evaluation, the extent of dissection was classified according to the modified DeBakey classification [29], and in combination with the clinical presentation, the hemodynamic status was assessed using the Penn classification [30].

While there are individual differences in the specific operative procedures, most surgeries for AAD at our center follow a previously described standard methodology [31,32]. Briefly, patients were transferred to the operating theater immediately upon diagnosis or admission, where standard monitoring (including at least two arterial pressure lines, core body temperature measurement, cerebral oximetry, and transesophageal echocardiography) was established. While cannulation strategies changed over time, after the initiation of cardiopulmonary bypass (CBP), the patient was cooled to the desired target temperature. After aortic cross-clamping and cardiac arrest following the administration of cardioplegia, the aortic valve and root were inspected and treated with repair or replacement strategies as required. The ascending aorta was replaced, and in select cases with limited dissection, distal anastomosis was performed at the tangentially clamped aorta. In hemiarch procedures, distal anastomosis is performed at the level of the brachiocephalic trunk, including the inner curvature of the arch. If complete arch replacement using the frozen elephant trunk (FET) technique was used, the head vessels were reimplanted separately after the distal anastomosis was completed. Both hemiarch and arch procedures were generally performed in moderate hypothermic circulatory arrest (HCA) with selective cerebral perfusion. The patients were then rewarmed, weaned from CPB, and admitted to our intensive care unit (ICU) for further treatment and monitoring.

### 2.3. Statistical Analysis

Statistical analysis was carried out using IBM SPSS Statistics version 27 (IBM, Armonk, NY, USA), Wizard Pro version 1.9.7 (Evan Miller, Chicago, IL, USA), and GraphPad Prism version 10 (GraphPad Software, Boston, MA, USA). For the comparison of the four groups, the chi-square test, Fisher’s exact test, and analysis of variance (ANOVA) were used, as appropriate. Long-term survival was assessed by Kaplan–Meier curves created with GraphPad Prism and compared using the log-rank test. All statistical tests were 2-sided, the alpha level was set at 0.05 for statistical significance, and a 95% confidence interval (CI). Continuous data are presented as mean ± standard deviation, or, if the mean was not skewed due to outliers, as median (interquartile range), frequency data as absolute numbers (percentage), survival length as median survival [95% CI], and survival at specific time points as a percentage [95% CI]. Statistically significant *p* values are denoted in bold.

## 3. Results

### 3.1. Patient Demographics and Presurgical Clinical Presentation

During the study period, 912 patients with a BMI of 18.5 or greater were operated on for AAD, consisting of 582 (63.8%) male and 330 (36.2%) female patients (Table 1). The mean age was 64.5 ± 13.6 years, and common comorbidities included arterial hypertension (72.3%), smoking history (21.5%), and coronary artery disease (18.0%). The normal weight group (BMI 18.5–24.9) consisted of 332 patients, the overweight group (BMI 25–29.9) of 367 patients, the obesity class I group (BMI 30–34.9) of 146 patients, and the obesity class II+ group (BMI ≥ 35) consisted of 67 patients.

While the sex distribution was skewed toward the male sex in all groups, the proportion of male patients was significantly increased in overweight and obese patients (*p* < 0.001). The average age decreased from BMI 25–29.9 (66.0 ± 12.7 years) to BMI 30–34.9 (62.6 ± 12.5 years) and BMI ≥ 35 (59.7 ± 12.5 years) groups. In the study cohort, patients with a higher BMI also had a greater body surface area (BSA). Patients in the higher BMI groups showed a significantly higher prevalence of arterial hypertension (*p* = 0.005) and diabetes mellitus (*p* < 0.001).

The median follow-up duration was 4.4 [3.9–4.8] years, with a 97% complete follow-up and no statistically significant difference between groups, although patients with BMI ≥ 35 tended to have a shorter duration of follow-up.

Most patients (70.6%) presented with the more extensive DeBakey Type I dissection, without significant differences between the different BMI groups (Table 2). A similar number of patients (69.0%) had aortic valve regurgitation preoperatively. Generalized malperfusion (circulatory shock) occurred in 23.9% of patients, and cardiopulmonary resuscitation was required in 8.9% of patients. At the time of admission, 13.8% of the patients were already intubated and ventilated. These findings were similar in all BMI groups.

Among patients with localized malperfusion (41.2%), the most common localization was cerebral malperfusion (15.8%), followed by peripheral (14.9%) and coronary (12.9%) malperfusion. While the fraction of patients with any localized malperfusion was comparable between the four groups (40.1–42.5%, *p* = 0.947), the distribution of the affected organ system showed some variations with mesenteric (*p* = 0.003), and—to a lesser extent—renal (*p* = 0.071) and peripheral (*p* = 0.291) malperfusion being more common in the higher BMI groups. Patients with BMI ≥ 35 had a lower rate of cerebral malperfusion, and neurological deficits preoperatively compared to the other groups. However, these findings did not reach statistical significance.

The ischemic burden, consisting of both generalized and localized malperfusion, was assessed using the Penn classification. Without reaching statistical significance, patients in the obesity II group (BMI ≥ 35) showed a slightly higher proportion of isolated localized malperfusion (Penn B) and a slightly lower proportion of both general and localized malperfusion (Penn BC).

### 3.2. Operative Details

The majority of patients (depending on the group between 50.7% and 58.7%) were treated with replacement of the ascending aorta and hemiarch replacement (Table 3). This was followed by the isolated proximal repair (25.9–34.9%) and replacement of ascending aorta and aortic arch (14.4–19.4%). Overall, the aortic repair strategy was similar in all groups.

The aortic valve was reconstructed in 69.7% of the patients. While the rate of aortic valve repair increased from normal weight (BMI 18.5–24.9; 67.8%) to obesity class I (BMI 30–34.9; 74.0%), patients with obesity class II or higher had the lowest rate of aortic valve repair (BMI ≥ 35; 55.2%). Aortic valve replacement was performed in 10.2% of patients, with overweight patients (BMI 25–29.9; 7.6%) requiring the lowest rate of aortic valve replacement.

Other concomitant procedures like root replacement and coronary artery bypass grafting (CABG) did not show statistically significant differences. The duration of CPB, aortic cross-clamping, and HCA increased with higher BMI; however, these differences were not statistically significant (*p* = 0.446, *p* = 0.682, and *p* = 0.225, respectively). The lowest temperature during HCA was 23.2 ± 5.6 °C without differences between groups, corresponding to low moderate hypothermia [15].

As shown in Table 4, patients in the BMI ≥ 35 group had renal failure, requiring dialysis almost twice as often as those in the other groups (*p* = 0.010). While tracheotomy was also more frequently performed in these patients, and their hospital stay was slightly longer, these differences were not statistically significant. Furthermore, the duration of postoperative ventilation increased with higher BMI; however, these findings did not reach statistical significance.

The in-hospital mortality was lowest in normal weight patients (BMI 18.5–24.9: 10.8%) and highest in obese patients (BMI 30–34.9: 15.8%, BMI ≥ 35: 14.9%, *p* = 0.461, Table 5). The long-term survival during the observed follow-up was similar in all groups (*p* = 0.894, Figure 1).

## 4. Discussion

We presented a retrospective analysis of 912 patients surgically treated for AAD stratified according to their BMI into four groups (normal weight, overweight, obesity class I, obesity class II+). To our knowledge, this constitutes the largest single-center study evaluating the effect of obesity on long-term survival in patients with AAD.

The patient demographics, comorbidities, and distribution of the Penn classification in the study cohort match previously reported characteristics of patients with AAD [33,34,35,36,37,38,39,40]. Therefore, the present study cohort is representative of patients with AAD, and the study’s conclusions should be applicable to the general population of patients with AAD.

When comparing patient demographics across these four groups, it was apparent that obese patients were younger than overweight and normal weight patients. While further studies are necessary, these results suggest that an increased BMI constitutes a risk factor for AAD, leading to the condition presenting earlier in life in patients with obesity.

This might be conferred by the higher rates of chronic arterial hypertension associated with obesity [1,2,41,42,43,44]. In the present study, obese patients suffered from arterial hypertension or diabetes mellitus significantly more often despite their younger age. A similar but statistically not significant observation was made for smoking history.

The shifted sex distribution toward the male sex in all groups corroborated the knowledge of the male sex being a risk factor for AAD [45]. Higher BMI groups had a significantly higher male patient proportion compared to the normal weight group. This further supports the conclusion that a higher BMI might be an AAD risk factor, with male patients with a higher BMI possibly being at even higher risk than male patients with a lower BMI.

Since heavier patients in our cohort tended to be younger, this might explain why, despite obesity being an established risk factor for CAD [41,46], there was no difference in CAD prevalence between our groups. Obesity is a known COPD risk factor [47], which might explain the tendency for higher COPD prevalence in the BMI 30–34.9 group. Another explanation might be the slightly higher prevalence of smoking in this group. The lower prevalence of COPD in the BMI ≥ 35 group, which did not reach statistical significance, might be incidental and should be researched in a larger patient population.

The Penn classification describes the total ischemic burden of the organism and has been demonstrated to correlate with early mortality [39,48]. While the organ system affected by localized malperfusion showed some variations, the overall presence of any localized malperfusion was similar in all four groups. Likewise, the distribution of Penn classes did not reveal significant differences. This suggests that BMI is not a major factor contributing to preoperative localized or generalized malperfusion.

There was also no difference in most other preoperative parameters. Notable, preoperative CPR, aortic regurgitation, DeBakey Type, and the share of patients with previous cardiac surgery were similar between the four groups. The significantly higher mesenteric malperfusion might also be an incidental finding, as other studies did not make the same observation [17].

The surgical approach did not differ significantly between the different BMI groups. Patients with a higher BMI had slightly longer CPB, aortic cross-clamping, and HCA durations and a higher temperature without statistical significance. Given the higher volume-to-surface ratio, it is conceivable that patients with a higher BMI are more challenging to cool down to the desired core body temperature for HCA. This is most likely the reason why patients with higher BMI have longer CBP and aortic cross-clamping times than leaner patients.

There were no significant differences in the short- or long-term survival between the patient groups. However, it should be noted that obese patients were younger, and therefore the remaining statistical life expectancy was higher. This was not accounted for in our analysis. Notably, a paradoxically better survival for overweight or obese patients was not observed.

There are only a few comparative studies that investigated the outcome of AAD repair surgery based on a patient’s BMI. Some authors have found higher early mortality for overweight or obese patients [17,21,25,49], but the comparability to our study collective is limited due to different surgical approaches and regional differences. Most studies with a longer follow-up did not find significant differences in long-term survival [17,18,22]. This is in line with our results and confirms that overweight or obese patients have a similar long-term life expectancy after AAD surgery.

Some studies have described a longer hospital stay or ventilation time in obese patients [22,23]. In our study, the hospital stay and time to extubation were not significantly different between the four patient groups, although median ventilation time was slightly longer in patients with a higher BMI. However, the comparability is again limited due to possible regional differences between the patient populations of Kreibich et al. (Philadelphia, PA, USA), Liu et al. (Beijing, China), and the present study (Mainz, Germany). Earlier remobilization, which is more common in the USA, might decrease typical complications of immobilized morbidly obese patients (e.g., pneumonia, decubitus) that prolong the hospital stay. Furthermore, the surgical approach was different, with Lui et al. only analyzing patients treated with FET.

In conclusion, our results suggest that for the outcome of AAD repair surgery, no obesity paradox that predicts better outcomes for overweight or obese patients exists. These findings are supported by those of previous research.

Nevertheless, the long-term mortality in obese and morbidly obese patients may be positively influenced by the fact that these patients are, on average, younger than leaner patients when they develop AAD. Additional research is necessary to fully understand the influence of body composition on the outcome after AAD surgery.

### Limitations

The significance of our findings is limited by the retrospective design of this study. Further parameters that might be valuable in the setting of obese patients to assess body composition (e.g., waist-to-hip ratio and fat mass measurements) or cardiopulmonary fitness (e.g., pulmonary function tests, VO_2_ max measurements) were not available due to the necessary immediate surgical treatment. Postoperative measurements are not routinely performed, as they would introduce a survival bias and might not correctly reflect the patient’s state at admission. However, body composition can be estimated from CTs, which are generally available for almost all patients with AAD [50,51]. This could be a subject of future research.

The number of obese patients was lower (obesity class I: 146 patients, obesity class II+: 67 patients) compared to that in the normal weight (332 patients) and overweight (367 patients) groups.

## 5. Conclusions

Diabetes and hypertension are known comorbidities of obesity and could be responsible for the comparatively early-in-life occurrence of aortic dissections in overweight and obese patients. A higher BMI appears to constitute a risk factor for AAD.

No differences in short- or long-term survival with regard to BMI were found, but this might be influenced by the longer remaining statistical life expectancies of patients with higher BMI.

The overall favorable outcome of surgery for all patient groups in our study highlights that immediate surgical therapy and close postoperative follow-up are essential for the treatment of patients with acute Type A aortic dissection, regardless of their body weight.

## Figures and Tables

**Figure 1 life-14-00955-f001:**
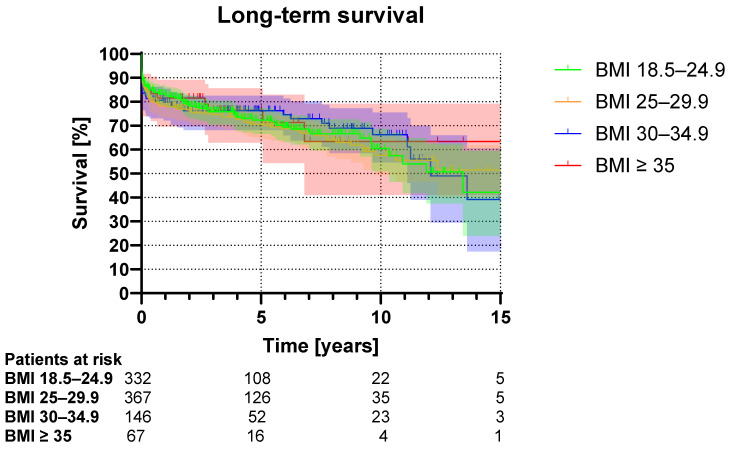
Kaplan–Meier curve illustrating the long-term survival. Shaded areas indicate 95% confidence interval.

**Table 1 life-14-00955-t001:** Patient demographics.

Variable	Total	BMI 18.5–24.9	BMI 25–29.9	BMI 30–34.9	BMI ≥ 35	*p* Value
	(*n* = 912)	(*n* = 332)	(*n* = 367)	(*n* = 146)	(*n* = 67)	
Sex						**<0.001**
Male	582 (63.8%)	184 (55.4%)	251 (68.4%)	102 (69.9%)	45 (67.2%)	
Female	330 (36.2%)	148 (44.6%)	116 (31.6%)	44 (30.1%)	22 (32.8%)	
Age [years]	64.5 ± 13.6	64.7 ± 14.9	66.0 ± 12.7	62.6 ± 12.5	59.7 ± 12.5	**0.001**
BMI [kg/m^2^]	27.5 ± 5.1	23.2 ± 1.5	27.3 ± 1.3	31.9 ± 1.3	40.2 ± 5.9	**<0.001**
BSA [m^2^]	1.99 ± 0.25	1.82 ± 0.18	2.01 ± 0.17	2.16 ± 0.19	2.37 ± 0.27	**<0.001**
Comorbidities						
Arterial Hypertension	659 (72.3%)	222 (66.9%)	266 (72.5%)	114 (78.1%)	57 (85.1%)	**0.005**
Diabetes mellitus	83 (9.1%)	16 (4.8%)	32 (8.7%)	19 (13.0%)	16 (23.9%)	**<0.001**
Smoking history	196 (21.5%)	67 (20.2%)	74 (20.2%)	39 (26.7%)	16 (23.9%)	0.347
CAD	164 (18.0%)	58 (17.5%)	70 (19.1%)	28 (19.2%)	8 (11.9%)	0.541
COPD	83 (9.1%)	25 (7.5%)	37 (10.1%)	18 (12.3%)	3 (4.5%)	0.173
Previous cardiac surgery	63 (6.9%)	23 (6.9%)	28 (7.6%)	10 (6.8%)	2 (3.0%)	0.639
Median follow-up [years]	4.4 [3.9–4.8]	4.1 [3.5–4.7]	4.7 [3.9–5.5]	4.8 [3.7–5.9]	3.5 [2.6–4.3]	0.116

BMI: body mass index; BSA: body surface area; CAD: coronary artery disease; COPD: chronic obstructive pulmonary disease. Statistically significant *p* values are denoted in bold.

**Table 2 life-14-00955-t002:** Preoperative status.

Variable	Total	BMI 18.5–24.9	BMI 25–29.9	BMI 30–34.9	BMI ≥ 35	*p* Value
	(*n* = 912)	(*n* = 332)	(*n* = 367)	(*n* = 146)	(*n* = 67)	
DeBakey classification						0.512
Type I	644 (70.6%)	228 (68.7%)	259 (70.6%)	150 (71.9%)	52 (77.6%)	
Type II	268 (29.4%)	104 (31.3%)	108 (29.4%)	41 (28.1%)	15 (22.4%)	
Aortic valve						
Regurgitation	629 (69.0%)	228 (68.7%)	254 (69.2%)	103 (70.5%)	44 (65.7%)	0.911
Bicuspid valve	37 (4.1%)	17 (5.1%)	13 (3.5%)	5 (3.4%)	2 (3.0%)	0.723
CPR	81 (8.9%)	35 (10.5%)	28 (7.6%)	15 (10.3%)	3 (4.5%)	0.284
Shock	218 (23.9%)	75 (22.6%)	91 (24.8%)	41 (28.1%)	11 (16.4%)	0.268
True lumen collapse	217 (23.8%)	72 (21.7%)	81 (22.1%)	41 (28.1%)	23 (34.3%)	0.072
Intubated/Ventilated	126 (13.8%)	41 (12.3%)	53 (14.4%)	27 (18.5%)	5 (7.5%)	0.129
Malperfusion	376 (41.2%)	1439 (41.9%)	147 (40.1%)	62 (42.5%)	28 (41.8%)	0.947
Coronary	118 (12.9%)	549 (14.8%)	39 (10.6%)	22 (15.1%)	8 (11.9%)	0.339
Cerebral	144 (15.8%)	58 (17.5%)	64 (17.4%)	17 (11.6%)	5 (7.5%)	0.077
Spinal	21 (2.3%)	7 (2.1%)	10 (2.7%)	2 (1.4%)	2 (3.0%)	0.751
Renal	89 (9.8%)	24 (7.2%)	35 (9.5%)	20 (13.7%)	10 (14.9%)	0.071
Mesenteric	106 (11.6%)	34 (10.2%)	33 (9.0%)	24 (16.4%)	15 (22.4%)	**0.003**
Peripheral	136 (14.9%)	44 (13.3%)	52 (14.2%)	26 (17.8%)	14 (20.9%)	0.291
Penn classification						0.423
A	445 (48.8%)	168 (50.6%)	180 (49.0%)	64 (43.8%)	33 (49.3%)	
B	249 (27.3%)	89 (26.8%)	96 (26.2%)	41 (28.1%)	23 (34.3%)	
C	91 (10.0%)	25 (7.5%)	40 (10.9%)	20 (13.7%)	6 (9.0%)	
BC	127 (13.9%)	50 (15.1%)	51 (13.9%)	21 (14.4%)	5 (7.5%)	
Pericardial effusion						0.633
Pericardial effusion	393 (43.1%)	149 (44.9%)	146 (39.8%)	66 (45.2%)	32 (47.8%)	
Tamponade	147 (16.1%)	49 (14.8%)	61 (16.6%)	27 (18.5%)	10 (14.9%)	
Neurological status						0.284
No neurologic deficits	714 (78.3%)	260 (78.3%)	282 (76.8%)	113 (77.4%)	59 (88.1%)	
Neurologic deficits	147 (16.1%)	54 (16.6%)	66 (18.0%)	21 (14.4%)	5 (7.5%)	
Not obtainable	51 (5.6%)	17 (5.1%)	19 (5.2%)	12 (8.2%)	3 (4.5%)	

CPR: cardiopulmonary resuscitation. Statistically significant *p* values are denoted in bold.

**Table 3 life-14-00955-t003:** Operative details.

Variable	Total	BMI 18.5–24.9	BMI 25–29.9	BMI 30–34.9	BMI ≥ 35	*p* Value
	(*n* = 912)	(*n* = 332)	(*n* = 367)	(*n* = 146)	(*n* = 67)	
Aortic Repair						0.535
Isolated proximal repair	262 (28.7%)	86 (25.9%)	106 (28.9%)	51 (34.9%)	19 (28.4%)	
Ascending, hemiarch	510 (55.9%)	195 (58.7%)	206 (56.1%)	74 (50.7%)	35 (52.2%)	
Ascending, arch	140 (15.4%)	51 (15.4%)	55 (15.0%)	21 (14.4%)	13 (19.4%)	
Concomitant procedures						
Aortic valve						**0.025**
Repair	636 (69.7%)	225 (67.8%)	266 (72.5%)	108 (74.0%)	37 (55.2%)	
Replacement	93 (10.2%)	438 (11.4%)	28 (7.6%)	18 (12.3%)	9 (13.4%)	
Root replacement	82 (9.0%)	39 (11.7%)	23 (6.3%)	13 (8.9%)	7 (10.4%)	0.086
CABG	118 (12.9%)	46 (13.9%)	47 (12.8%)	16 (11.0%)	9 (13.4%)	0.855
Perfusion details						
CPB [min]	187 ± 87	184 ± 80	189 ± 86	186 ± 99	202 ± 97	0.446
Cross-clamping [min]	102 ± 48	102 ± 46	100 ± 44	104 ± 58	106 ± 49	0.682
HCA [min]	25.1 ± 12.2	24.9 ± 11.5	24.7 ± 11.8	25.1 ± 13.2	28.9 ± 15.1	0.225
Lowest temperature [°C]	23.2 ± 5.6	22.7 ± 5.6	23.4 ± 5.4	23.9 ± 5.7	23.5 ± 6.3	0.119

CABG: coronary artery bypass grafting; CPB: cardiopulmonary bypass; HCA: hypothermic circulatory arrest. Statistically significant *p* values are denoted in bold.

**Table 4 life-14-00955-t004:** Postoperative course.

Variable	Total	BMI 18.5–24.9	BMI 25–29.9	BMI 30–34.9	BMI ≥ 35	*p* Value
	(*n* = 912)	(*n* = 332)	(*n* = 367)	(*n* = 146)	(*n* = 67)	
Rethoracotomy	107 (11.7%)	42 (12.7%)	44 (12.0%)	14 (9.6%)	7 (10.4%)	0.790
Tracheotomy	61 (6.7%)	20 (6.0%)	28 (7.6%)	6 (4.1%)	7 (10.4%)	0.263
Dialysis	147 (16.1%)	42 (12.7%)	65 (17.7%)	21 (14.4%)	19 (28.4%)	**0.010**
Ventilation [hours]	21.0 (52.5)	19.0 (48.0)	22.0 (54.3)	24.0 (55.5)	26.0 (97.0)	0.345
Hospital stay [days]	14.1 ± 12.2	13.7 ± 8.3	14.5 ± 10.8	12.7 ± 9.4	16.2 ± 29.1	0.202
Neurological status						0.058
No neurologic deficits	702 (77.0%)	251 (75.6%)	283 (77.1%)	116 (79.5%)	52 (77.6%)	
Neurologic deficits	139 (15.2%)	59 (17.8%)	59 (16.1%)	16 (11.0%)	5 (7.5%)	
Not obtainable	71 (7.8%)	22 (6.6%)	25 (6.8%)	14 (9.6%)	10 (14.9%)	

Statistically significant *p* values are denoted in bold.

**Table 5 life-14-00955-t005:** Short- and long-term survival.

Variable	Total	BMI 18.5–24.9	BMI 25–29.9	BMI 30–34.9	BMI ≥ 35	*p* Value
	(*n* = 912)	(*n* = 332)	(*n* = 367)	(*n* = 146)	(*n* = 67)	
In-hospital mortality	115 (12.6%)	36 (10.8%)	46 (12.5%)	23 (15.8%)	10 (14.9%)	0.461
Median survival	13.4 [10.8–16.0]	13.4 [9.7–17.2]	15.4 [8.7–22.1]	12.1 [9.5–14.7]	n/a *	0.894
1-Year Survival	80.6% [77.8–83.0%]	83.2% [78.6–86.9%]	78.3% [73.7–82.3%]	79.8% [72.2–85.4%]	81.5% [69.7–89.1%]	
5-Year Survival	72.6% [69.2–75.6%]	72.4% [66.5–77.4%]	70.7% [65.1–75.5%]	76.3% [68.2–82.6%]	76.4% [62.9–85.6%]	
10-Year Survival	60.5% [55.3–65.2%]	60.5% [51.0–68.7%]	57.6% [49.3–65.0%]	66.2% [54.6–75.5%]	63.4% [41.1–79.2%]	
15-Year-Survival	46.4% [36.9–55.4%]	42.2% [24.0–59.5%]	51.6% [40.2–61.8%]	39.2% [17.5–60.5%]	63.4% [41.1–79.2%]	

* As the survival during the observed follow-up in this cohort did not reach levels less than 50%, the median survival could not be calculated.

## Data Availability

The datasets presented in this article are not readily available because of privacy restrictions. Reasonable requests to access the datasets should be directed to the corresponding author.

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
