# Peer review of "The Effect of Obesity on Short- and Long-Term Outcome after Surgical Treatment for Acute Type A Aortic Dissection"

_life, 2024, doi:10.3390/life14080955_

Round 1

Reviewer 1 Report

Comments and Suggestions for Authors

The authors aimed to evaluate the effect of obesity on short- and long-term outcomes after surgical treatment for acute Type A aortic dissection (AAD)

The authors found that obese patients are affected from AAD at a younger age, and therefore obesity might constitute a risk factor for AAD. They found also that obesity does not influence short- or long-term survival. 

I think this study is novel and very important as it fills also a gap in the current literature.

Methods are robust and well described.

The introduction is too short and needs more background information.

Results are very relevant.

Discussion is sometime out of track. It should focused on the main topics of the article with a more critical point of view, especially in the light of other similar articles in the current literature.

Author Response

Comment 1: The authors aimed to evaluate the effect of obesity on short- and long-term outcomes after surgical treatment for acute Type A aortic dissection (AAD)

The authors found that obese patients are affected from AAD at a younger age, and therefore obesity might constitute a risk factor for AAD. They found also that obesity does not influence short- or long-term survival.

I think this study is novel and very important as it fills also a gap in the current literature.

Methods are robust and well described.

Response 1: Thank you for reviewing the article, and the encouraging feedback and valid concern. We aimed to address the raised points in the revision.

Comment 2: The introduction is too short and needs more background information.

Response 2: We have expanded the introduction and included more information about current research (page 2, line 93). We hope this fulfills your request.

Comment 3: Results are very relevant.

Discussion is sometime out of track. It should focused on the main topics of the article with a more critical point of view, especially in the light of other similar articles in the current literature.

Response 3: We have condensed the discussion of our results and focused on the main points of our manuscript (page 8, line 1139). The comparison to previous studies has been restructured to focus on the comparison of the primary endpoints. (page 9, line 1230).

Reviewer 2 Report

Comments and Suggestions for Authors

The authors report an original retrospective chart-review study on the association of acute type A aortic dissection surgery with complications over two time-points (though it is not clear what discrete amount of time separated short- from long-term) for all-comers at a single center stratified by BMI. Though the article certainly bears merit, a few fundamental concerns are immediately apparent and necessitate clarification to allow for a full evaluation to be completed:

1. It is stated that "A paradox of lower morbidity and mortality in obese patients undergoing cardiac surgery has been described" in the abstract, however upon reviewing the cited literature (references 15-18), this paradox is not clearly appreciated by the reviewer. Only one study with a conflicting conclusion is appreciated, suggesting the the literature supports obesity being associated with worse outcomes. Please note the exact quotes from the references below:

a) "In this meta-analysis, we observed a positive association between BMI and perioperative mortality of ATAAD through integration of 12 independent studies (Graphic Abstract)."

Song W, Liu J, Tu G, Pan L, Hong Y, Qin L, Wei L, Chen J. Impact of body mass index on perioperative mortality of acute stanford type A aortic dissection: a systematic review and meta-analysis. BMC Cardiovasc Disord. 2023 Oct 31;23(1):531. doi: 10.1186/s12872-023-03517-z. PMID: 37907847; PMCID: PMC10617194.

b) "Despite the comorbidities that are associated with obesity, obese patients undergoing surgical repair of ADA are not at greater risk of death or other adverse outcomes."

Kreibich M, Rylski B, Bavaria JE, Branchetti E, Dohle D, Moeller P, Vallabhajosyula P, Szeto WY, Desai ND. Outcome After Operation for Aortic Dissection Type A in Morbidly Obese Patients. Ann Thorac Surg. 2018 Aug;106(2):491-497. doi: 10.1016/j.athoracsur.2018.03.035. Epub 2018 Apr 16. PMID: 29673638.

c) "Obesity may increase these patients' operative risk; overweight does not."

Shimizu T, Kimura N, Mieno M, Hori D, Shiraishi M, Tashima Y, Yuri K, Itagaki R, Aizawa K, Kawahito K, Yamaguchi A. Effects of Obesity on Outcomes of Acute Type A Aortic Dissection Repair in Japan. Circ Rep. 2020 Oct 23;2(11):639-647. doi: 10.1253/circrep.CR-20-0098. PMID: 33693190; PMCID: PMC7937495.

d) "Patients with obesity (BMI, ≥30 kg/m2) who underwent surgery for type A acute aortic dissection had higher operative mortality rates and an increased risk of low cardiac output syndrome, pulmonary complications, and other postoperative morbidities than did patients without obesity."

Lio A, Bovio E, Nicolò F, Saitto G, Scafuri A, Bassano C, Chiariello L, Ruvolo G. Influence of Body Mass Index on Outcomes of Patients Undergoing Surgery for Acute Aortic Dissection: A Propensity-Matched Analysis. Tex Heart Inst J. 2019 Feb 1;46(1):7-13. doi: 10.14503/THIJ-17-6365. PMID: 30833831; PMCID: PMC6378991.

As such, the premise of the entire paper will likely need to be revised as no concrete paradox can be appreciated by this reviewer.

2. The authors are likely aware that body mass index is a useful however tarnished metric for comparing body habitus due to the fact that body composition can vary widely and height-weight combinations make for nearly infinite permutations. For example, a short (relative to average height for sex) sedentary male with relatively low muscle mass may have a similar BMI to an average height athletic female with relatively high muscle mass.

a) Perhaps this limitations should be addressed in the article, in addition to other limitations regarding BMI.

b) Are any other sources of information available to better stratify these patients within a BMI group, perhaps as a sub-group analysis? This may assist the authors with reporting accurate results that are less subject to confounding

c) What measures would the authors recommend be performed (or what data should be collected) during future aortic surgeries to allow for a future study to properly stratify patients into homogenous groups to allow for high-quality, rigorous, and reproducible work?

3. Was any data regarding cardiorespiratory fitness available to improve the stratification of patients and subsequent correlation analyses?

4. The authors state that "Further parameters that might be valuable in the setting of obese patients (e. g. waist circumference), were not available due to their missing routine measurement during the hospital stay" and presumably the authors are referring to the higher quality metric of waist-to-hip ratio. However it is difficult to imagine that of the 924 patients, none had any additional biometric information available.

Perhaps the authors could identify the minority of the cohort that has such biometric data available and perform a subsequent analysis, to improve the findings of the present article and shed light on a novel finding (improving this publication's priority and justifying its utility in the literature)?

Author Response

Comment 1: The authors report an original retrospective chart-review study on the association of acute type A aortic dissection surgery with complications over two time-points (though it is not clear what discrete amount of time separated short- from long-term) for all-comers at a single center stratified by BMI. Though the article certainly bears merit, a few fundamental concerns are immediately apparent and necessitate clarification to allow for a full evaluation to be completed

Response 1: Thank you for the feedback. We used the available clinical follow-up to assess long-term survival and used the in-hospital mortality to describe short-term survival. The in-hospital mortality is mentioned in many studies concerning AAD surgery and allows comparability. We have added a definition in the Material and Methods section (Study design and patients, page 2, line 122).

Comment 2: It is stated that "A paradox of lower morbidity and mortality in obese patients undergoing cardiac surgery has been described" in the abstract, however upon reviewing the cited literature (references 15-18), this paradox is not clearly appreciated by the reviewer. Only one study with a conflicting conclusion is appreciated, suggesting the the literature supports obesity being associated with worse outcomes. Please note the exact quotes from the references below:

  1. a) "In this meta-analysis, we observed a positive association between BMI and perioperative mortality of ATAAD through integration of 12 independent studies (Graphic Abstract)."

Song W, Liu J, Tu G, Pan L, Hong Y, Qin L, Wei L, Chen J. Impact of body mass index on perioperative mortality of acute stanford type A aortic dissection: a systematic review and meta-analysis. BMC Cardiovasc Disord. 2023 Oct 31;23(1):531. doi: 10.1186/s12872-023-03517-z. PMID: 37907847; PMCID: PMC10617194.

  1. b) "Despite the comorbidities that are associated with obesity, obese patients undergoing surgical repair of ADA are not at greater risk of death or other adverse outcomes."

Kreibich M, Rylski B, Bavaria JE, Branchetti E, Dohle D, Moeller P, Vallabhajosyula P, Szeto WY, Desai ND. Outcome After Operation for Aortic Dissection Type A in Morbidly Obese Patients. Ann Thorac Surg. 2018 Aug;106(2):491-497. doi: 10.1016/j.athoracsur.2018.03.035. Epub 2018 Apr 16. PMID: 29673638.

  1. c) "Obesity may increase these patients' operative risk; overweight does not."

Shimizu T, Kimura N, Mieno M, Hori D, Shiraishi M, Tashima Y, Yuri K, Itagaki R, Aizawa K, Kawahito K, Yamaguchi A. Effects of Obesity on Outcomes of Acute Type A Aortic Dissection Repair in Japan. Circ Rep. 2020 Oct 23;2(11):639-647. doi: 10.1253/circrep.CR-20-0098. PMID: 33693190; PMCID: PMC7937495.

  1. d) "Patients with obesity (BMI, ≥30 kg/m2) who underwent surgery for type A acute aortic dissection had higher operative mortality rates and an increased risk of low cardiac output syndrome, pulmonary complications, and other postoperative morbidities than did patients without obesity."

 Lio A, Bovio E, Nicolò F, Saitto G, Scafuri A, Bassano C, Chiariello L, Ruvolo G. Influence of Body Mass Index on Outcomes of Patients Undergoing Surgery for Acute Aortic Dissection: A Propensity-Matched Analysis. Tex Heart Inst J. 2019 Feb 1;46(1):7-13. doi: 10.14503/THIJ-17-6365. PMID: 30833831; PMCID: PMC6378991.

As such, the premise of the entire paper will likely need to be revised as no concrete paradox can be appreciated by this reviewer.

Response 2: The quoted sentence was only referring to cardiac surgery, not AAD surgery. The obesity paradox has been described (see sources 1-11) for patients with congestive heart failure and for patients undergoing cardiac surgery (CABG) and TAVR. In the context of AAD, the published studies have yielded different results. We clarified this in the abstract (page 1, line 14) and expanded the respective paragraph in the introduction (page 1, line 42) to better present the reader the current state of research.

Comment 3: The authors are likely aware that body mass index is a useful however tarnished metric for comparing body habitus due to the fact that body composition can vary widely and height-weight combinations make for nearly infinite permutations. For example, a short (relative to average height for sex) sedentary male with relatively low muscle mass may have a similar BMI to an average height athletic female with relatively high muscle mass.

  1. a) Perhaps this limitations should be addressed in the article, in addition to other limitations regarding BMI.

Response 3: We have updated the limitations and added possible influence of different body compositions (page 10, line 1328). Unfortunately, we do not believe it is possible to routinely perform traditional body composition measurements preoperatively.

Comment 4: b) Are any other sources of information available to better stratify these patients within a BMI group, perhaps as a sub-group analysis? This may assist the authors with reporting accurate results that are less subject to confounding

Response 4: Unfortunately, no parameters regarding body composition or cardiorespiratory fitness were available for a representative number of patients. Sex and age are other factors to consider, which showed significant differences between the four groups. When looking only at one sex, there were still no significant differences in the in-hospital mortality or long-term survival, but we did not include this in the manuscript as we believe the sex comparison is not the focus of this manuscript. The younger age of obese patients is discussed (page 9, line 1226) and we interpreted this to conclude that obesity might constitute a risk factor for AAD (page 8, line 1140).

Comment 5: c) What measures would the authors recommend be performed (or what data should be collected) during future aortic surgeries to allow for a future study to properly stratify patients into homogenous groups to allow for high-quality, rigorous, and reproducible work?

Response 5: As the surgery must be conducted as soon as possible, it is difficult to obtain additional preoperative data, and postoperative collection induces a survival bias. However body composition can be estimated from preoperative CTs, which are available for almost all patients. This analysis is beyond the scope of this manuscript, but it has been added to the discussion (page 10, line 1333) as a suggestion for further research.

Comment 6: 3. Was any data regarding cardiorespiratory fitness available to improve the stratification of patients and subsequent correlation analyses?

Response 6: Unfortunately, relevant parameters to assess cardiorespiratory fitness (e. g. pulmonary function test, VO2 max measurements, or cardiac stress test) were not broadly available. The medical history did not provide these parameters from recent assessments in a representative number of patients. Preoperative measurements were not possible due to the necessary urgent surgical treatment. Postoperative measurements of these parameters are not routinely performed in our clinic and would introduce a survivor bias. Furthermore, postoperative measurements would be limited in their significance due to the possible reduction of cardiorespiratory fitness due the dissection and/or perioperative complications.

Comment 7: The authors state that "Further parameters that might be valuable in the setting of obese patients (e. g. waist circumference), were not available due to their missing routine measurement during the hospital stay" and presumably the authors are referring to the higher quality metric of waist-to-hip ratio. However it is difficult to imagine that of the 924 patients, none had any additional biometric information available.

Perhaps the authors could identify the minority of the cohort that has such biometric data available and perform a subsequent analysis, to improve the findings of the present article and shed light on a novel finding (improving this publication's priority and justifying its utility in the literature)?

Response 7: As a university hospital, many patients did not present to our hospital prior to the diagnosis of AAD. Furthermore, parameters regarding body composition and cardiorespiratory fitness are generally obtained in the outpatient setting, e. g. by a general practitioner. These findings were not available to our hospital and were therefore not saved in our digital clinical archive. While it might be possible to obtain some data, this would require additional patient cooperation and consent to share this data for research and additional ethical review. Since our study period dates back to 2004, it might not be possible or feasible to obtain this data from general practitioners for earlier cases. Our goal in this manuscript was to analyze the available data for all patients, as subgroup analysis introduces a bias we desired to avoid. However, as noted in Response 5, the available CTs might be used to assess body composition. In our opinion this might be more feasible and available for more patients than the gathering of traditional body composition measurements from external healthcare providers.

Reviewer 3 Report

Comments and Suggestions for Authors

Authors should be congratulated for their work but methods and results should be revised to support conclusions and to improve the quality of the manuscript. 

1. "hospital stay was slightly longer" is not supported by analysis, please remove from abstract (P=0.178)

2. Can the results be grouped according to "non obese" (BMI < 30) vs "obese" (BMI >= 30)? The low sample size of very obese patients might have reduced the power of the analysis considering the 4-groups comparison and postestimation tests. Postestimation tests are not included in the tables and analysis, and this is relevant. I would suggest to repeat the analysis, using only 2 groups (non-obese vs obese). 

3. Patients with low BMI (< 25) were still included in the study. Should this be revised? Considering the study hypothesis, BMI < 25 should be an exclusion criteria to reduce potential bias, and those 7 patients (lines 126-127) should be excluded.

4. Survival (Table) from Kaplan Meier analysis should include confidence intervals.

5. Kaplan Meier figures should include confidence intervals and censored cases (i.e. hash mark on survival function).

Author Response

Comment 0: Authors should be congratulated for their work but methods and results should be revised to support conclusions and to improve the quality of the manuscript.

Response 0: Thank you very much for the detailed review and feedback. We have reviewed our statistical analysis accordingly and hope this address your concerns.

Comment 1: "hospital stay was slightly longer" is not supported by analysis, please remove from abstract (P=0.178)

Response 1: Thank you for pointing this out, we have removed this sentence from the abstract (page 1, line 24).

Comment 2: Can the results be grouped according to "non obese" (BMI < 30) vs "obese" (BMI >= 30)? The low sample size of very obese patients might have reduced the power of the analysis considering the 4-groups comparison and postestimation tests. Postestimation tests are not included in the tables and analysis, and this is relevant. I would suggest to repeat the analysis, using only 2 groups (non-obese vs obese).

Response 2: We considered this in the design of the study as it provides the benefits you mentioned. We opted to split the cohort into 4 groups (according to overweight and obesity definitions) to assess whether any group has a better outcome. As mentioned in the introduction, in some cases overweight (BMI 25-30), but not obese patients, have the best outcome, and this cannot be assessed using only two groups. Furthermore, our approach allows comparability to previously published studies. We understand that the statistical analysis is more robust and more likely to yield relevant significant results with only two groups, but when performing the suggested grouping (BMI cutoff 30) we found no significant differences in the short and long-term survival.

Comment 3: Patients with low BMI (< 25) were still included in the study. Should this be revised? Considering the study hypothesis, BMI < 25 should be an exclusion criteria to reduce potential bias, and those 7 patients (lines 126-127) should be excluded.

Response 3: I believe you refer to underweight patients with BMI < 18.5. Thank you for pointing this out, it is of course correct to exclude underweight patients. We have updated the study to exclude these 12 patients (using 18.5 as a cutoff) and recalculated our statistical analysis, without causing significant changes in our results (page 2, line 115).

Response 4: 95% CI has been added to Table 5 (page 7, line 203).

Comment 5: Kaplan Meier figures should include confidence intervals and censored cases (i.e. hash mark on survival function)

Response 5: 95% CI have been added as shaded areas in Figure 1 (page 8, line 206). The censored cases were already marked with an upward tick. The legend has been corrected and the tick size has been increased to improve visibility.

Round 2

Reviewer 2 Report

Comments and Suggestions for Authors

The reviewer thanks the authors for the detailed responses. While the content was clearly described, the corresponding manuscript text changes were not highlighted/red-lined. Thus this reviewer is unable to appreciate what changes have been made the manuscript. The reviewer requests that the authors perform this standard courtesy to enable prior peer review.

Author Response

Comment 1: The reviewer thanks the authors for the detailed responses. While the content was clearly described, the corresponding manuscript text changes were not highlighted/red-lined. Thus this reviewer is unable to appreciate what changes have been made the manuscript. The reviewer requests that the authors perform this standard courtesy to enable prior peer review.

Response 1: Please excuse the missing highlighting in the uploaded version. I have uploaded an updated version with all changes clearly marked.

Reviewer 3 Report

Comments and Suggestions for Authors

Authors should be congratulated for the revised version of the manuscript. I have no specific comment, and manuscript should be considered for publication, in my opinion.

Author Response

Thank you for the review and feedback.

Round 3

Reviewer 2 Report

Comments and Suggestions for Authors

The reviewer thanks the author(s) for their point-by-point responses and manuscript revisions.  Only one minor concern now remains:

1. What is meant by "Schock" in Table 2, row 9?

Author Response

Comment 0: The reviewer thanks the author(s) for their point-by-point responses and manuscript revisions.  Only one minor concern now remains:

Response 0: Thank you for the review.

Comment 1: What is meant by "Schock" in Table 2, row 9?

Response 1: Thank you for pointing this out, the typo has been corrected. We refered to preoperative circulatory shock. If preoperative vital signs were available, the shock index was calculated. Otherwise, shock was assessed using the available documentation, including from emergency personnel, external hospitals, and preoperative recorded values.